# Tissue-Predisposition to Cancer Driver Mutations

**DOI:** 10.3390/cells13020106

**Published:** 2024-01-05

**Authors:** Luriano Peters, Avanthika Venkatachalam, Yinon Ben-Neriah

**Affiliations:** Lautenberg Center for Immunology and Cancer Research, Institute for Medical Research (IMRIC), The Faculty of Medicine, Hebrew University of Jerusalem, P.O. Box 12272, Jerusalem 91120, Israel; luriano.peters@mail.huji.ac.il (L.P.); avanthik.venkatachal@mail.huji.ac.il (A.V.)

**Keywords:** cancer evolution, driver mutations, selective pressure

## Abstract

Driver mutations are considered the cornerstone of cancer initiation. They are defined as mutations that convey a competitive fitness advantage, and hence, their mutation frequency in premalignant tissue is expected to exceed the basal mutation rate. In old terms, that translates to “the survival of the fittest” and implies that a selective process underlies the frequency of cancer driver mutations. In that sense, each tissue is its own niche that creates a molecular selective pressure that may favor the propagation of a mutation or not. At the heart of this stands one of the biggest riddles in cancer biology: the tissue-predisposition to cancer driver mutations. The frequency of cancer driver mutations among tissues is non-uniform: for instance, mutations in *APC* are particularly frequent in colorectal cancer, and 99% of chronic myeloid leukemia patients harbor the driver *BCR-ABL1* fusion mutation, which is rarely found in solid tumors. Here, we provide a mechanistic framework that aims to explain how tissue-specific features, ranging from epigenetic underpinnings to the expression of viral transposable elements, establish a molecular basis for selecting cancer driver mutations in a tissue-specific manner.

## 1. Introduction

While rediscovering Mendel’s genetics, the term “mutation” was coined by Hugo de Vries, in 1901, to explain the apparent differences of progeny from its direct parent generation. Mutationists like de Vries did not reject evolution but believed that it takes place in large and sudden leaps which are entirely driven by genetic events. In contrast, Charles Darwin’s gradualism assumed that small variations occur throughout a long period of time, and the driving force is external factors leading to natural selection [1]. Natural selection, however, positively selects beneficial traits and may explain the survival of the fittest but not how the fittest was created. Darwin’s theory was therefore insufficient to explain the ongoing variation within a population [2]. Nevertheless, even without the *Modern Synthesis* [3]—a modern interpretation encompassing Darwin’s and Mendelian genetics—a cancer biologist would reconcile both hypotheses and fuse them based on their experimental experience, as many aspects of cancer follow both Darwin’s and de Vries’ assumptions. Mutations may bring along constant and instant phenotypic changes that natural selection can act on. This evolutionary concept diverges experimentally into two scientific questions: first, how do genetic variations occur in the first place, and second, how are these variations selected for? While the last decades have shed light on several mechanisms of how and when mutation(s) occur [4]—for example, defects in DNA repair mechanisms or the overproduction of reactive oxygen species—the selective pressures underlying clonal evolution are less well understood. Above all stands one of the biggest riddles in cancer biology: the tissue-predisposition to cancer driver mutations.

The great advancement in research technology has been able to uncover a spectrum of phenomena, one of which is that mutations in certain genes show a frequency bias in certain tissues for no apparent reason. The best example is germline mutations in *BRCA1*, which predispose female patients to developing ovarian cancer or breast cancer [5]. While all cells have this mutation, only a couple of tissues seem to be affected. Similarly, germline mutations in *APC* lead to Familial Adenomatous Polyposis (FAP), a condition that may develop into a malignancy mostly in the colon [6] (Figure 1A). Likewise, somatic mutations in *TP53* are the most common mutations in cancer, but they show a clear frequency bias towards certain tissues [7], namely, the lung, pancreas, colon, ovaries, and the esophagus. Additionally, over 90% of pancreatic cancer patients harbor mutations in *KRAS* [8], but much less in other tissues. Probably the most extreme example is the fusion of the genes *ABL1* and *BCR*, which seems to be entirely exclusive to hematological malignancies [9] (Figure 1B).

The fact that the aforementioned mutated genes are rather regulators of cell autonomous biological processes than cell type-specific biomarkers means that a tissue-predisposition to cancer driver mutations may stem from certain shared features that other tissues lack. These features could collectively result in the imposition of selective pressures for certain mutations. Here, we provide a mechanistic framework that aims to explain how tissue-specific features, ranging from epigenetic underpinnings to the expression of viral transposable elements, form a molecular basis for selecting cancer driver mutations in a tissue-specific manner.

## 2. Lessons from Normal Tissue

### 2.1. Mutations Are Encoded by the Epigenetic Landscape

Cell identity is the result of a delicate interplay between cell type-specific transcription factors and chromatin modifying enzymes that maintain an epigenetic landscape which is permissive to the functional needs of a cell. As such, the epigenetic landscape is the fingerprint of functionally distinct cells [10,11]. DNA is tightly wrapped around nucleosomes, the units of which are histones, which form the first layer of structural organization in our genome. How tightly DNA is wrapped around nucleosomes can be regulated by chemical modification of histones’ residues and stands in direct correlation with the regulation of gene expression. Chromatin is further organized into topological associated domains (TADs), in which patches of chromatin are spatially separated from others to allow for the spatiotemporal regulation of genes [12,13]. As early as the late 1980s, researchers had already discovered that mutation rates within our genome are non-uniform and can be associated with the genetic composition of a gene [14]. It was only much later that concrete evidence emerged, demonstrating the connection between the structural organization of our genome and the diverse mutation rates that are present throughout it. In particular, it was acknowledged that the three-fold methylation of the ninth residue in histone 3 (H3K9me3) could explain nearly half of the variations in mutation rates in cancer cells [15]. Promptly following this discovery, researchers began identifying additional epigenetic characteristics, including nucleosome occupancy and various other histone modifications, that contributed to the understanding of the variation in mutation rates [16]. The next stage of investigations focused on understanding whether the cell type-specific epigenetic profile of healthy cells could predict tissue-specific cancer driver mutations. Polak et al. pursued this hypothesis and initially demonstrated that DNA hypersensitivity assays, a measure of chromatin accessibility in melanocytes, could account for the mutation pattern that is observed in melanoma. They also found that the histone mark H3K4me1 in melanocytes explained the regional mutation rate in melanoma, although this relationship did not hold true for other types of cancer [17]. Notably, mutation rates are elevated in regions of closed chromatin, aligning with the observation that the repressive mark H3K9me3 accounted for a significant portion of the mutation rate variability throughout the genome [15]. Accordingly, Polak et al. also confirmed a negative correlation of histone marks that are related to active or open chromatin (H3K27ac, H3K4me1) in normal cells to the mutation rate in their cognate cancer (Figure 2). Using the most informative epigenetic features, Polak et al. developed an algorithm that was able to predict the cancer cell of origin based on the mutational pattern in several cancers.

Investigations into the cell of origin are of particular value when studied together with tissue metaplasia. Metaplastic cells undergo a transient change in cell identity, accompanied by epigenetic changes, which usually is a consequence of environmental cues, such as cigarette smoke [18]. Metaplasia is often the initial cellular and histological change associated with cancer progression. Interestingly, it was shown that epigenetic features of metaplastic cells in the esophagus are better predictors of the mutational pattern in esophageal cancer than those of non-metaplastic cells in the same tissue [19]. These investigations marked the initial instance where the strong connection between the epigenetic environment in both healthy and premalignant cells and the mutational pattern in cancer was revealed. They underscored the notion that the tissue-specific mutational profile might already be predetermined by the epigenetic characteristics of the cell’s origin.

### 2.2. Gene Expression or Function

Much as the epigenetic landscape serves as a fingerprint of a cell’s identity, gene expression levels could also contribute to cell identity and are the most reliable approach to identify distinct cell types [20,21]. Consequently, it is a fair assumption that cell type-specific gene signatures in normal cells may explain tissue-specific mutations in cancer. However, the expression level of a gene is not a direct measure of its impact. For instance, transcription factors usually show lower expression levels, but their impact on cell fate decisions is substantial. Indeed, gene expression in normal cells alone is a weak predictor for tissue-specific mutations in cancer compared to the epigenetic landscape [17], and if it does predict these, the correlation tends to be negative [22], and this may be largely attributed to a process called transcription-coupled DNA repair. To ensure that cell function is maintained properly, several mechanisms to secure genome integrity have evolved specifically around genes that are essential for the cell’s survival. Transcription-coupled DNA repair ensures that mutations encountered during transcription will be repaired “on the fly” and is mediated through the pathways of nucleotide and base excision repair [23]. This notion is in line with observations that mutation rates in cancer cells are higher in areas of closed chromatin of their normal counterparts (see section above). However, a paradox arises from the fact that the gene expression level in normal cells is not as good a predictor of mutation rates as is closed chromatin. This discrepancy might arise because correlations do not consider the timing of mutations during cancer progression and the cellular state at which the cell was while mutations took place (e.g., activated or metaplastic). It rather seems that in normal cells, the expression level of genes acting in the mutation pathway is a better predictor of mutation rates [24]. This finding has major implications, suggesting that biological processes, and hence the function, in the cell of origin (which may be cell type-specific) are guiding the occurrence of tissue-specific mutations.

### 2.3. Developmental Origins of Cancer

In mammals, during embryonic development, various signaling pathways, such as the *BMP/TGF-β*, *NOTCH*, *WNT/β-catenin*, Hedgehog, and Hippo pathways, intricately govern cell division, growth, and differentiation to regulate development within each lineage [25]. Even after undergoing differentiation, these cells preserve this memory in the form of epigenetic imprints, which are reflected in DNA methylation patterns and histone marks [26]. This preexisting epigenetic landscape in different tissues determines how oncogenes and tumor suppressor genes can influence a tissue response to a specific oncogenic signal, which is contingent on the permissiveness of its epigenetic state [27]. Consequently, different cell types can react to a particular stimulus, such as an oncogenic mutation, either similarly, entirely differently, or in non-responsive ways.

In vertebrates, the Notch pathway is crucial to post-lineage segregation, and its inactivation leads to premature neuronal progenitor differentiation, indicating its role in maintaining a progenitor state and inhibiting differentiation [28]. Specific Notch pathway elements and downstream effectors are expressed in the developing pancreas, and this mechanism is reactivated in *KRAS*-mutant acinar cells, initiating the *EGFR/SOX9* axis [29]. Subsequently, this axis restores the progenitor traits induced by Notch signaling while inhibiting the differentiation of metaplastic cells, forming the platform for pancreatic ductal adenocarcinoma (PDAC) development. Interestingly, the *EGFR/SOX9* axis is maintained through the *PI3K/AKT* and not the MAPK pathway, as the deletion of *CRAF* has no further effect on PanIN formation, but mutations in *PI3K* drive acinar to ductal metaplasia on its own [30]. These findings highlight the reactivation of Notch signaling downstream of *KRAS*, favoring the pathway that is able to maintain the axis, and hence determining which mutations may effectively establish it. The high prevalence of *KRAS* mutations, but not mutations in other MAPK components, in PDAC may therefore be a consequence of its ability to perturb tissue homeostasis most effectively by affecting Notch signaling in the pancreas.

Notch signaling is not only an integral part of pancreatic cell fate decision and its tissue homeostasis but is also a major determinant of such in the hematopoietic system, where active Notch signaling mediates hematopoietic cell fate determination in the embryo and is a critical factor in the maintenance of self-renewing hematopoietic stem cell (HSC) pools and in T-lineage cell differentiation, including T versus B cells, in adults [31]. When *NOTCH1* is conditionally deleted in hematopoietic progenitors, it leads to a halt in early T-cell development and the buildup of abnormal immature B-cells within the thymus [32,33]. Despite the ubiquitous expression of the Notch receptor throughout the hematolymphoid compartment, Notch activation induces transformation primarily in developing T cells. A classic example of a Notch-related cancer is human acute T-cell acute lymphoblastic leukemia/lymphoma (T-ALL), accounting for around 15–20% of cases in both children and adults [34]. In addition, recent research showed that continuous Notch signaling in malignant murine and human B cells has unveiled widespread Notch-mediated growth inhibition and apoptosis in both immature and mature B cell malignancies [35]. The cell type-specific effects observed in closely related cells underscore the significance of mutations and the cell of origin. When Notch functions as a guardian of stem cells or a regulator of the precursor cell fate, it fulfils an oncogenic role, whereas its tumor-suppressing activity becomes evident in tissues where Notch signaling triggers terminal differentiation events.

The Wnt-β-catenin-signaling pathway is another evolutionarily conserved communication system that governs many stages of the development, and particularly, the fate of endodermal cells gives rise to the linings of two tubes within the body [36]. The first tube extends throughout the length of the body, forming the digestive tube, and the differentiated buds from this tube form the liver, gallbladder, and pancreas. Wnt ligands are released by Wnt-producing cells, exerting their influence over nearby Wnt-responsive cells across varying distances. This process is continued in the adult intestine, one of the most rapidly self-renewing tissues in the body, and is essential throughout life. Within the intestine, *WNT3* and *WNT9B* are expressed within Paneth cells, while *WNT2B* and *WNT5A* originate from Foxl1-positve mesenchymal cells, thereby activating Wnt signaling in both stem cells and transit amplifying cells [37]. As Wnt activation happens only in the proximity of Wnt ligand-secreting cells, in the intestinal stem cell niche, the dividing cells start to differentiate as they move away from the base of crypts. Thus, most tissues employ a specific regulatory mechanism to promote differentiation and limit stem cell expansion, which could be hijacked to initiate tumorigenesis [38]. In the case of sporadic colorectal cancers, mutational inactivation of both *APC* alleles leads to Wnt hyperactivation, and these cells gain the ability to proliferate independently of the intestinal stem cell niche and form neoplastic lesions [39]. As mentioned earlier, a hereditary cancer syndrome known as Familial Adenomatous Polyposis (FAP) results from a germline *APC* mutation. FAP patients inherit one faulty *APC* allele and have normal embryonic development but develop numerous adenoma polyps only in the colon string in early adulthood [40]. Ultimately, a subset of these polyps transforms into malignant adenocarcinomas. These polyps and adenomas are benign and characterized by the inactivation of the second *APC* allele, and this occurs only in colonic stem cells and not in other highly proliferative cells, like hematopoietic stem cells, highlighting the hijacking of developmental mechanisms that maintain tissue homeostasis [41]. An alternate hypothesis, that is also rooted in developmental origins, is that serrated polyps arise from metaplasia, as opposed to stem cell expansion, giving rise to colon tumors. This hypothesis proposes that serrated polyps originate from mature cells through a “top-down” model of tumorigenesis, wherein genes that are typically present in the gastric pylorus are re-expressed [42]. This re-expression results in a reversion to a fetal gene program and a loss of regional identity, accompanied by diminished *CDX2* expression, a common feature observed in serrated tumors. This Wnt-independent tumorigenesis is triggered by MAPK activation through Braf-activating mutations, epithelial damage response, or stress resulting from a deficiency in mismatch repair [43]. However, *BRAF* mutations in mouse models require a “second hit” event, such as the disruption of transforming growth factor-β (TGF-β) signaling, to induce tumorigenesis. This “second hit” may arise from signals within the microenvironment, or by the loss of *CDX2* expression. All together, these studies highlight that understanding the relationship between the cell of origin and its developmental trajectory and homeostasis is a valuable mechanistic insight into the occurrence of tissue-specific driver mutations.

## 3. The Molecular Context of Mutations in Cancerous Tissue

### 3.1. The Paralogue Paradox

Tissue-specific characteristics of cancers are a rule, not an anomaly. The robustness in cell-specific signaling is attained through controlled buffering mechanisms, through genetic redundancy and compensatory pathways [44]. Genetic redundancy denotes the phenomenon where two or more genes exhibit either complete or partial functional compensation for each other in the same pathway. Paralogues are a good explanation for such compensatory effects, but their mutation rate is not uniform among cancers, which is paradoxical, as one would expect that functionally similar paralogues can also compensate for each other regarding their cancer-driving properties. A study investigating cell type-specific mutations in various RAS family proteins in mice explains why human tumors exhibit varying frequencies of *KRAS*, *HRAS*, and *NRAS* mutations across different types [45]. *KRAS* stands out as the predominant mutation in non-small cell lung cancer (NSCLC), while *HRAS* mutations are prevalent in skin cancer. To test if the cancer-specific mutation rate of either paralogue depends on their genomic locus, one might try to replace *KRAS* by *HRAS*. By utilizing an advanced knock-in technique to express wild-type *HRAS* from the native *KRAS* locus in mice, it was shown that *HRAS* codon 61 mutations specifically occurred in NSCLC only when expressed from the *KRAS* locus and never occurred in the original *HRAS* locus [45]. Hence, the low mutation rates of *HRAS* in NSCLC could be explained by its genomic locus. These results emphasize that the unique mechanisms behind *KRAS* mutations in NSCLC and *HRAS* mutations in skin cancer are based on tissue-specific gene-regulatory elements rather than inherent differences in the functionality of the encoded proteins.

### 3.2. Synthetic Lethality

Cancer cells have a “conditional dependence” on their paralog partners or compensatory partners, and if we therapeutically or genetically inactivate the partner, it will selectively affect cancer cells with the inactivated gene, sparing normal cells and leading to synthetic lethality [46]. In addition, compensatory mechanisms involve components from multiple signaling pathways and represent nodes that control oncogenic pathways via negative feedback loops. This provides a substantial explanation for the occurrence of mutual exclusivity, which refers to the absence of two or more commonly observed mutations in individual tumors within specific cancer types. Striking examples are *KRAS* and *EGFR* proto-oncogenes in human lung adenocarcinomas, *NRAS* and *BRAF* in melanoma, and *PTEN* and *PIK3CA* in sarcoma [47]. However, synthetic lethality is not restricted to targets within the same pathway, but cancer driver mutations show a remarkable dependence on components of other pathways in a tissue-specific manner. In a comprehensive study utilizing RNA-interference screens across various cell lines to investigate which genes, when silenced, kill cells bearing only mutant *RAS*, *TBK1* emerged as the most effective choice upon therapeutic *KRAS* inhibition [48]. *TBK1* acts as an upstream regulator of the NF-κB pathway by phosphorylating and inactivating IκB, which inhibits NF-κB, indicating that NF-κB activation is necessary to promote Ras-tumorigenesis. In another instance, mice with mutant KRAS and deletion of TP53 in lung tissue developed highly aggressive lung cancer, but the concurrent expression of the same NF-κB pathway inhibitor, IκB “super repressor”, significantly reduced both tumor count and size [49]. This effect appears to be tissue-specific, as in a mouse model of *RAS*-driven human skin cancer, NF-κB inhibition led to tumor progression, possibly due to the role of NF-κB in a tumor’s immunogenicity [50]. These studies demonstrate that exploring synthetic lethality is an effective approach to identifying tissue-specific co-dependencies of cancer drivers. Tissue-specific patterns of tumor suppressor genes can also be rationalized by considering the expression levels of genes whose disruption would trigger a synthetic lethal effect. For example, the incidence of *BRCA1* mutations is significantly elevated in breast and ovarian cancers, which coincide with an elevated expression of *GSTM5* in these normal tissues when compared to organs like the liver and kidney [51]. Thus, reduced expression of *GSTM5* could result in an adverse selective pressure for *BRCA1* mutations (Figure 3A). Together, the concept of synthetic lethality serves not only in the identification of therapeutic vulnerabilities, but astonishingly as an explanation for the tissue-specific occurrence of cancer driver mutations.

### 3.3. To Deal or Not to Deal with Senescence

Cellular senescence is mediated by several stimuli such as oncogenic stress, genomic instability, and telomer shorting, wherein cells exit the cell cycle terminally even when being exposed to mitogenic stimuli. In sharp contrast, the term “quiescence” was long reserved for cells residing in G0, yet still being capable of re-entering the cell cycle after sensing mitogens [52,53]. While the p53/p21 and the p16/Rb pathways have been identified as the major routes for cellular senescence, there are a plethora of non-canonical intermediate signals that depend not only on the genetic lesion but seem to be cell type-specific [54]. Senescence-associated phenotypes are therefore expected to differ among tissues. More than a decade ago, we have shown that the deletion of *CK1α* in the intestine induces senescence accompanied by a senescence-associated inflammatory response (SIR) that is protumorigenic but remarkably distinct from the commonly appreciated senescence-associated secretory phenotype (SASP) [55]. Different phenotypic outcomes observed in senescent states may originate from distinctive cell type responses to specific genetic lesions. These could determine whether and how the cell undergoes senescence. For example, germline *BRCA1* haploinsufficiency poses a major risk of developing breast and ovarian cancer. While all cells in the breast tissue carry this mutation, specifically mammary epithelial cells undergo premature senescence in response to increased genomic instability and telomer erosion, but not mammary fibroblasts of the same genotype. Unexpectedly, *BRCA1* haploinsufficiency-induced premature senescence of mammary epithelial cells is independent of p53 and p16, but associated with increased levels of Rb acetylation, which are mediated through decreased expression levels of *SIRT1*, a negative regulator of Rb acetylation [56]. This was in stark contrast to the fibroblasts tested, which underwent senescence later than mammary epithelial cells and with activation of p53 and p16. Moreover, it has been shown that only *BRCA1* mutant basal and stromal cells, but not mature luminal or luminal progenitors, showed distinguished epigenetic profiles (among them H3K27ac and H3K4me3) compared to non-carriers [57]. Although these findings were observed ex vivo, they impressively demonstrate that different primary cell types carrying the same genetic lesion can present diverse responses rather than a uniform and predictable response, explaining why different cell fates may be variably determined by similar mutations. To support this view, Falcomatà et al. generated mouse models in which Cre is expressed under the *PDX1* promoter, a transcription factor that is expressed in progenitor cells of the ventral foregut endoderm and involved in the specification of both the extrahepatic bile duct and the pancreas [58]. Introducing the knock-in mutations *KRAS^G12D^*, a driver of >90% of PDAC, and *PI3K^H1047R^*, a driver of extrahepatic cholangiocarcinoma (ECC), in the common progenitor of both tissues allows for the exploration of tissue-specific effects of these mutations. Interestingly, *PI3K^H1047R^* is able to induce pancreatic intraepithelial neoplasia (PanIN) and biliary intraepithelial neoplasia (BiIIN), while *KRAS^G12D^* only induces PanIN. This directly shows in vivo that different cell types have diverse responses to oncogenic signaling, if at all. In fact, *KRAS^G12D^* was not able to induce BiIIN or ECC even within 800 days. Remarkably, *PI3K^H1047R^* induced p53-independent senescence in the extrahepatic bile duct but not in the pancreas, and upon deletion of *P27KIP1*, progression toward ECC was eased. Strikingly, the deletion of *P27KIP1* favored the development of ECC on the background of *KRAS^G12D^*, emphasizing that *P27KIP1* is a gate-specific tumor suppressive in the extrahepatic bile duct. In contrast, the ablation of *TP53* on both oncogenic backgrounds leads to tumor progression and PDAC, confirming that *TP53* is a major tumor-suppressive checkpoint specifically to overcome OIS in the pancreas [59]. The latter finding is crucial to the understanding that early oncogenic driver mutations can drive cancer in both tissues, but their ability to do so depends on a cell type-specific inactivation of the tumor suppressor response (Figure 3B). 

### 3.4. LINE-1 Transposable Elements

Most of the human genome consists of repetitive sequences that are known as transposable elements (TEs), among which long interspersed element-1 (*LINE-1*) accounts for most of the transposition events in humans. A bidirectional RNA polymerase II promoter can transcribe *LINE-1*, producing a transcript that includes two open reading frames (ORFs), ORF1 and ORF2, which encode for proteins (ORF1p, ORF2p) that are capable of mediating retrotransposition [60]. Although the expression of *LINE-1* was believed to be silenced in adult somatic cells by DNA methylation [61], there is evidence that suggests the expression of *LINE-1* in both developing and adult tissue [62,63]. The expression pattern of TEs in general is cell type-specific, and the chromatin accessibility information of TE loci was shown to be fully able to predict the cell type [64]. Less controversial is the aberrant re-expression of *LINE-1* during tumorigenesis, which is a common feature that varies dramatically between different cancer types [65], suggesting that cancers with deregulated *LINE-1* expression may be subject to an additional selective pressure that is very often overlooked. This may be because only a minority of *LINE-1* elements are retrotransposition-competent [66]; however, their ability to self-propagate may indicate that *LINE-1* retrotransposons are endogenous mutagens [63]. The best example is the *LINE-1* insertions into the *APC* locus leading to low expression levels of its gene product, ultimately driving colorectal cancer [67]. However, these events are rare, and their contribution to the occurrence of tissue-specific mutations is more likely a result of the disruption of gene-regulatory elements or biological processes.

The fact that *LINE-1* can autonomously propagate poses a significant genotoxic stress in cells that aberrantly re-express the retrotransposition-competent *LINE-1* as, for instance, its expression alone can induce DNA double-strand breaks, leading to apoptosis [68] and suggesting that tumor suppressive pathways are active in *LINE-1*-expressing cells. Supportive of that is that the expression of *LINE-1* in normal human fibroblasts triggers a senescence phenotype [63]. As pointed out above, the tumor-suppressive response may actively shape the occurrence of tissue-specific driver mutations, and therefore, the expression of *LINE-1* in only a subset of tissues is one determinant of that response. Accordingly, it was shown that the expression of ORF1p is higher in cancers with *TP53* mutations (Figure 4A), suggesting that *TP53* is a critical tumor suppressor gate to prevent *LINE-1*-induced genotoxic stress. Furthermore, the expression of *LINE-1* correlated with the activity of DDR components [69]. Tissues that show expression of *LINE-1* presumably have to deal with an ongoing DDR mediated through p53. That, in turn, might favor the occurrence of mutations in tumor suppressors. For example, in the early stages of esophageal cancer, the expression of ORF1p was detected in Barret’s esophagus [70], which poses a selective pressure that may favor the inactivation of *TP53*, the most common event in esophageal cancer (~65% of patients). Following that line of thought, other cancer types with a high frequency of *TP53* mutations (such as small cell lung cancer, pancreatic cancer, colorectal cancer, and ovarian cancer) may be those that most commonly show re-expression of *LINE-1*, which is indeed the case [60]. Regarding ovarian cancer, this mechanism may also serve as an additional explanation for the positive selection of *BRCA1/2* mutations. Conclusively, we must recognize *LINE-1* as an endogenous mutagen or as a yet another modulator of a cell type-specific tumor-suppressive response.

### 3.5. Replication Timing Dynamics

Replication timing is the spatiotemporal sequence of replication events. It implies that replication is a spatially discontinuous process on a continuous time scale. Hence, genomic regions may be replicated early or late during the process of the genome replication and are highly influenced by the structural organization of the genome [71]. Generally, euchromatin is replicated during the early S-phase and heterochromatin during the late S-phase [72]. The replication timing highly correlates with mutation rates, as increased mutation rates are associated with later replication events [17,22,73]. It has been shown that reduced diversity of nucleotides or the accumulation of single-stranded DNA (which is more susceptible to DNA damage) in later replicating regions are responsible for the higher mutations rates during late replication [73]. Additionally, DNA mismatch repair was shown to be more effective during early euchromatic replication events [74]. While mismatch repair provides another mechanistic explanation for the higher mutation rates in closed chromatin (in addition to transcription-coupled DNA repair), one needs to keep in mind that the epigenetic landscape dictating open and closed chromatin is a fingerprint that is unique to a cell. We may therefore assume that replication carries a cell type-specific spatiotemporal signature that can impact the tissue-specific occurrence of mutations. Indeed, replication timing is a conserved process, but at the same time, there is considerable timing variation in cell types of the same species. Ryba et al. showed that the replication time alignment rate of human embryonic stem cells to mouse epiblast-derived stem cells (mEpiSCs) is high, but the alignment of mEpiSCs to mouse embryonic stem cells was significantly lower [75]. Impressively, Caballero et al. showed that mutations rates correlate with the replication timing in a cell type-specific manner. Moreover, unique mutational signatures, such as G:T mismatches, stem from mutational processes (such as MMR defects) associated with late replication timing and show a clear cell type bias. For example, in chronic lymphocytic leukemia (CLL) cell lines, the mutation rate increases during late replication events, and the major mutational signature can be traced back to Polymerase eta (Pol η) defects. However, in a colorectal cancer (CRC) cell line, these mutational signatures associated with late replication are predominantly related to MMR defects [76]. These studies indicate that replication-timing-associated mutations may arise in a cell type-specific fashion, and additionally, that there is a cell type-specific bias of mutational processes during replication (Figure 4B). Intriguingly, the bias of mutational processes is conserved between normal cells and their respective cancerous counterparts [77]. Although this finding is counterintuitive, as transformation is accompanied by changes in the epigenetic landscape, it establishes the notion that a bias in mutational processes in cancer cells is linked to their respective normal cells of origin, suggesting that even early tissue-specific driver mutations can arise as a consequence of replication time dynamics.

## 4. Conclusions and Perspectives

Tissue-predisposition to cancer driver mutations represents one of the most enigmatic puzzles in the field of cancer biology. Despite significant advancements in research technology over the past decade, encompassing advanced sequencing techniques and sophisticated mouse models, the question of why cancer driver mutations exhibit tissue-specific patterns remains elusive. This phenomenon, revealed as part of our expanding knowledge of tumorigenesis, bears potential insights into the origins of cancer. In this review, our primary objective was to present a mechanistic framework that may uncover the molecular underpinnings behind tissue-predisposition to cancer driver mutations, aiming to push the scientific community to directly address this enigma.

A good example of tissue-predisposition to mutations are mutations in TP53. While germline mutations in *TP53* may predominantly cause cancers like sarcoma and lymphoma [78], somatic mutations predominantly affect tissues like the ovaries, pancreas, colon, and lungs (Figure 1). This observation highlights the fact that the timing, context, and kind of mutation is of the utmost importance to the understanding of tissue-specific driver mutations. The question is and remains what mechanisms determine this context. The choice of mutations is influenced by various biological processes that extend across entire realms of research. These processes, although individually studied to uncover novel therapeutic possibilities, are likely interconnected, forming a network of selective pressures that are specific to cell types. We have discussed how the epigenetic state is not only a major determinant of the tissue-specific occurrence of mutations [17], but also how it is directly linked to biological processes such as replication timing and tissue-specific gene expression [76]. Tumor-suppressive responses, such as OIS, are in turn linked to *LINE-1* expression [69]. Moreover, reactivation of developmental pathways is a hallmark of cancer, where the acquisition of immature features drives malignant transformation, and a unique set of mutations may be able to drive processes, such as EMT, in a tissue-specific manner. While unfolding possible intrinsic mechanisms behind the tissue bias of cancer driver mutations, there are also many other external factors that can shape this bias. An example is the skin, which is permanently exposed to UV light, an environmental stress sparing most other organs [79]. Addressing the effects of external factors where possible, while perturbing synthetically lethal targets, would be the most valuable strategy for elucidating the tissue bias.

Improvements in single-cell RNA sequencing and analysis may profoundly fill the gap in knowledge. Simultaneous measurement of the chromatin accessibility state and expression profiles in a cell type-specific manner [80], assisted by bioinformatic tools, can enable the exploration of clonal architectures based on single-cell RNA sequencing data [81]. This may reveal how and when mutations occur, while considering the relevant epigenetic states and driver genes’ expression levels. The explosive research on cancer driver mutations in normal tissue [82] will likely shed more light on tissue-specific cancer mutations. These “normal” mutant tissues are perfect biological systems for applying the technologies that are needed to understand the impact of confounding variables such as chromatin states and mutation rates in comparison to their non-mutant counterparts of the same normal cell type.

To conclude, tissue-predisposition to cancer driver mutations is subject to the molecular context that they occur in. This context, including intrinsic, genetic, and epigenetic, as well as extrinsic/environmental factors, could be revealed by screens aimed at perturbing synthetically lethal targets. Understanding the selective pressures that allow mutations to spread will also allow for the identification of therapeutic vulnerabilities. As responsiveness to a molecular alteration via mutation-targeted therapy frequently hinges on the tissue of origin [83,84], identifying the tissue-specific context of oncogenic mutations may have an important therapeutic value. Based on these considerations, the advancement of genomic analyses and perturbation tools may turn tissue-predisposition to cancer driver mutations into a non-random, predictable process.

## Figures and Tables

**Figure 1 cells-13-00106-f001:**
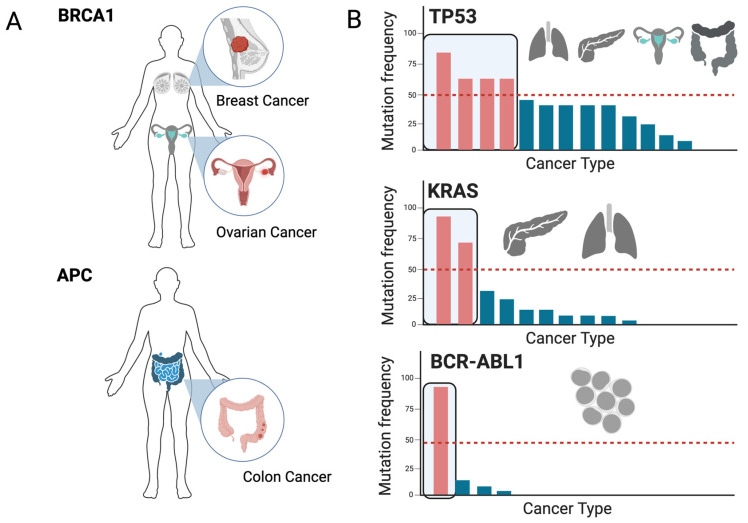
Tissue-predisposition to cancer driver mutations. (**A**) Germline mutations are present in all tissues of the body, but only specific tissues have an increased susceptibility to developing malignancies. For instance, germline mutations in *BRCA1* predispose female carriers to developing ovarian and breast cancer, while germline mutations in *APC* predispose carriers to developing colon cancer. (**B**) Somatic mutations in common regulators of cellular processes such as *KRAS*, *BCR*, and *TP53* are most commonly mutated in cancer, but the mutation frequency is biased towards certain tissues. Bar plots show the mutation frequency in several cancer types. Red bars show cancer types that have the highest frequency (>50%, red line), and the tissue symbols represent the respective tissues (the mutation frequencies are based on relevant datasets in cBioportal).

**Figure 2 cells-13-00106-f002:**
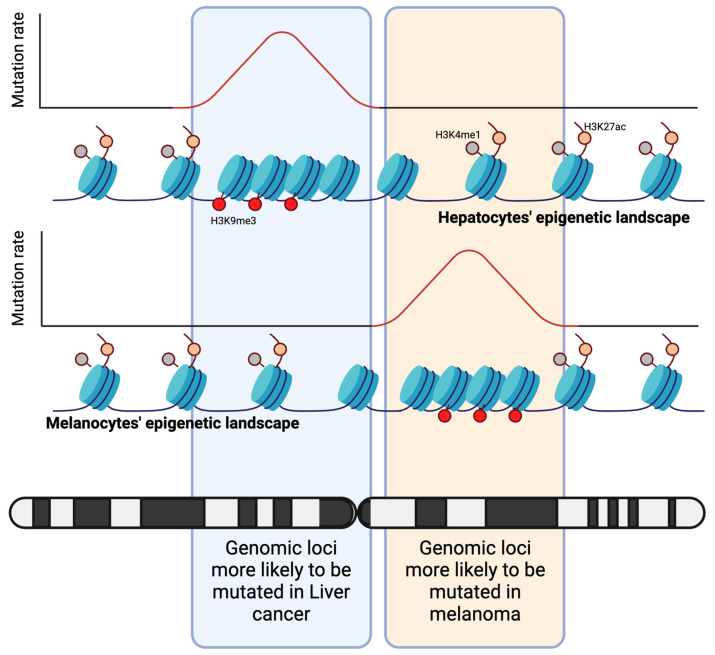
The epigenetic landscape of the cell of origin. The mutation rate is influenced by epigenetic attributes, such as histone marks. In normal cells like hepatocytes (upper chromatin sketch) and melanocytes (lower chromatin sketch), open chromatin indicated by H3K4me1 and H3K27ac exhibits lower mutation rates compared to closed chromatin marked by H3K9me3. Since their epigenetic profiles differ, the genomic loci more likely to undergo mutations that also vary in their respective cancers, as highlighted in the depicted chromatid regions.

**Figure 3 cells-13-00106-f003:**
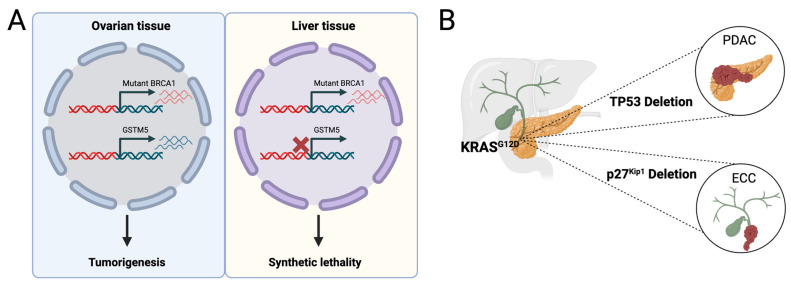
Biological processes affecting tissue bias of mutations. (**A**) *GSTM5* and *BRCA1* are synthetic lethality partners. A mutation in *BRCA1* therefore requires *GSTM5* expression. In ovarian tissue, the expression of *GSTM5* is high compared to liver tissue, which explains why *BRCA1* mutations have a higher tendency to propagate in the ovarian tissue. (**B**) Tumor suppressor gates are tissue-specific. Under *KRAS^G12D^*, the deletion of *CDKN1B* (p27^Kip1^) is required for the progression of ECC, and the deletion of *TP53* is required for the progression of PDAC.

**Figure 4 cells-13-00106-f004:**
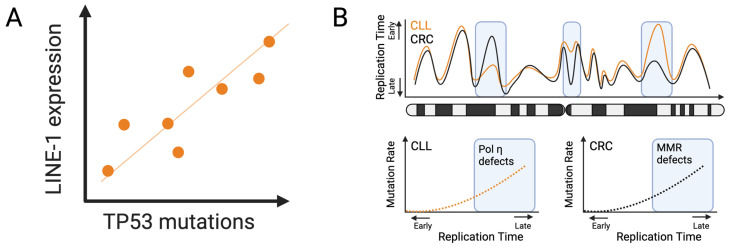
Determinants of tissue-specific mutation rates. (**A**) *LINE-1* expression correlates with *TP53* mutation rates. (**B**) Replication timing dynamics impact tissue-specific mutation rates in two ways: first, there are late and early replication events that are cell type-specific (orange line vs. black line in the upper panel); second, mutation-prone replication occurs in late replication events in general, but the mutational signature and the mutational processes are cell type-specific (lower panel). For example, in CLL cell lines, mutational signatures are based on Pol η defects, while in CRC cell lines, they are based on MMR defects.

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
