# Peer review of "Tissue-Predisposition to Cancer Driver Mutations"

_cells, 2024, doi:10.3390/cells13020106_

Round 1

Reviewer 1 Report

Comments and Suggestions for Authors

Review by Peter et al “Tissue-predisposition to cancer driver mutations” provides a unique and well referenced prospective on the relationship between tumorigenesis and mutations. Very well written review.

Minor points

Page 7 lines 259-261 sentence needs clarification and reference: “By utilizing an advanced knock-in technique to express wild-type HRAS from the native KRAS locus in mice, it was shown that HRAS codon 61 mutations specifically occurred in NSCLC only when expressed from the KRAS locus, and never occurred in the original HRAS locus (Ref).”

Figures need to be referenced in the text

Author Response

Dear Reviewer,

we appreciate your time to review our manuscript and we carefully went through your comments. Please find below a list of changes that we perfromed on the manuscript follwing the suggestions of all reviewers. 

We have mareked the corrections as suggested by Reviewer 1 in Blue, by Reviewer 2 in Yellow and the Author-corrections in Purple. 

We hope that the corrections are to your satisfaction. 

Reviewer 1

Line 49: Figure reference “(Fig 1A)”

Line 55: Figure reference “(Fig 1B)”

Line 106: Figure reference “(Fig 2)”

Line 255-261: as suggested by Reviewer, clarified the passage by changing it to “To test if the cancer-specific mutation rate of either paralogue depends on their genomic locus, one might try to replace KRAS by HRAS. By utilizing an advanced knock-in technique to express wild-type HRAS from the native KRAS locus in mice, it was shown that HRAS codon 61 mutations specifically occurred in NSCLC only when expressed from the KRAS locus, and never occurred in the original HRAS locus [45]. Hence, the low mutation rates of HRAS in NSCLC could be explained by its genomic locus.”

Line 296: Figure reference “(Fig 3A)”

Line 349: Figure reference “(Fig 3B)”

Line 391: Figure reference “(Fig 4A)”

Line 438: Figure reference “(Fig 4B)”

Reviewer 2 Report

Comments and Suggestions for Authors

The authors have submitted a well written and important review on the tissue specificity of cancer driver mutations. As the authors discuss, lack of a full mechanistic understanding of this phenomenon is a limitation to cancer research and with a better understanding will provide an avenue for improved therapeutic intervention in the future. The authors do a good job of reviewing the current evidence and theories that have provided insight to this area. The objective of the review is stated and the figures are clear and informative. The authors also provide interesting historical perspective during the introduction and relevant discussion of future directions in the conclusions. The article will add significantly to the special issue.

Below are minor grammatical edits and suggestions for consideration:

Line 16: Consider replacing “barely” with “rarely”

Line 39: …mutation(s)…

Line 45 Change “are” to “is”

Line 97: Change “in” to “on”

Line 157: …occurrence “of” tissue specific mutations

Line 207: …give rise “to” the linings of…

Line 224: Remove “of” (…but develop of numerous adenoma polyps…)

Line 262: Reference missing

Line: 268: May read better as “Cancer cells have a ‘conditional dependence’ on their paralog partners…

Line 274: Consider rewording “concurrent occurrence”

Line 375: Period after reference number instead of comma.

Line 439: “However” should be capitalized

Check location of definitions for abbreviations. For example, PDAC is used multiple times prior to its definition on line 349. Definitions for other abbreviations are missing.

The placement of Figure 3 at a point later in the manuscript may be more helpful. Its current location occurs before a large portion of the contents in the figure have been discussed. Splitting the figure up (A with B and C with D) may be a better option which would allow the figure contents to more closely accompany the information in the text.

Comments on the Quality of English Language

Minor adjustments. See comments.

Author Response

Dear Reviewer,

we appreciate your time to review our manuscript and we carefully went through your comments. Please find below a list of changes that we perfromed on the manuscript follwing the suggestions of all reviewers. 

We have mareked the corrections as suggested by Reviewer 1 in Blue, by Reviewer 2 in Yellow and the Author-corrections in Purple. 

We hope that the corrections are to your satisfaction. 

Reviewer 2

Line 16: as suggested by Reviewer, changed to “rarely”

Line 39: as suggested by Reviewer, changed to “mutation(s)”

Line 45: as suggested by Reviewer, changed to “is”

Line 95: as suggested by Reviewer, changed to “on”

Line 152: as suggested by Reviewer, added “of”

Line 172-173: as suggested by Reviewer, added “pancreatic ductal adenocarcinoma” to explain the abbreviation “PDAC”

Line 185-186: as suggested by Reviewer, added “hematopoietic stem cells” to explain the abbreviation “HSC”

Line 203: as suggested by Reviewer, added “to”

Line 220: removed “of”

Line 254-255: as suggested by Reviewer, added “Non-small cell lung cancer” to explain the abbreviation “NSCLC”

Line 267: as suggested by Reviewer, changed to “Cancer cells have an ‘conditional dependence’ on their paralog partners or compensatory partners”

Line 285: rephrased sentence as suggested by reviewer by deleting “concurrent occurrence”

Line 370: replaced comma by dot.

Line 434: capitalized the word “however” to “However”

As suggested by the Reviewer, Figure 3 was split into Figure 3 (old Figure 3 A&B) and Figure 4 (old Figure 3 C&D). The figures have been repositioned. Figure 3 is located in the beginning of page 9 and Figure 4 is located in the beginning of page 11.

Author-corrections

Throughout the manuscript: We changed the wording “tissue-bias” to “tissue-predisposition”. For example: Line 13 or Line 464

Line 4: Affiliation correction

Line 33: changed “sympathize with” to “reconcile both hypotheses”

Line 42: changed “bias” to “predisposition”

Line 57-60: changed “means that the tissue bias of cancer driver mutations may reside within common features that some tissue share while others don’t. These features may collectively translate into selective pressures acting on mutations.” to “means that tissue-predisposition of cancer driver mutations may stem from certain shared features that other tissues lack. These features could collectively result in the imposition of selective pressures for certain mutations.”

Line 73: added “on relevant datasets”

Line 109: changed “In that respect, tissue metaplasia is an interesting case.” to “Investigations into the cell of origin are of particular value when studied together with tissue metaplasia.”

Line 117: added “both”

Line 122-124: changed “Open chromatin, indicated by H3K4me1 and H3K27ac, exhibits lower mutation rates compared to closed chromatin marked by H3K9me3, in normal cells like hepatocytes (upper chromatin band) and melanocytes (lower chromatin band).” to “In normal cells like hepatocytes (upper chromatin sketch) and melanocytes (lower chromatin sketch), open chromatin indicated by H3K4me1 and H3K27ac, exhibits lower mutation rates compared to closed chromatin marked by H3K9me3.”

Line 129-131: changed “As much as the epigenetic landscape is a fingerprint of a cell’s identity, so is the expression levels of genes which will ultimately execute the functions required. The expression of certain genes is therefore a good approach to identify distinct cell types” to “Much as the epigenetic landscape serves as a fingerprint of a cell’s identity, gene expression levels could also contribute to cell identity and is the most reliable approach to identify distinct cell types”

Line 135: changed “dramatic” to “substantial”

Line 144-150: changed “However, the fact that gene expression levels in normal cells are weak predictors of mutation rates, but closed chromatin is, causes a paradoxical problem: one might expect driver mutation events to have a reasonable expression level, entailing related enhancers and gene promoters to present active histone marks. This discrepancy might originate from the fact that correlations do not consider when mutations occurred during cancer progression or in which cellular state a cell was (e.g., activated or metaplastic). Instead, it seems that in normal cells the expression level of genes acting in the same pathway of a gene with a tissue specific mutation in cancer is of predictive value [24].” to “However, a paradox arises from the fact that the gene expression level in normal cells is not as a good predictor of mutation rates, as is closed chromatin. This discrepancy might arise because correlations do not consider the timing of mutations during cancer progression and the cellular state at which the cell was while mutations took place (e.g., activated or metaplastic). It rather seems that in normal cells, the expression level of genes acting in the mutation pathway is a better predictor of mutation rates [24].”

Line 218: added “As mentioned earlier”

Line 227: changed “gives” to “giving”

Line 312-314: changed “The phenotype-discrepancy of senescent states may be explained if assuming that if and how a cell deals with senescence in response to a specific genetic lesion is an inherent and unique feature of a cell type.” to “Different phenotypic outcomes observed in senescent states may originate from distinctive cell type responses to specific genetic lesions. These could determine whether and how the cell undergoes senescence.”

Line 418-420: changed “As this is, mechanistically, another explanation (next to transcription- coupled DNA repair) for the observation that mutations rates in closed chromatin are higher” to “While mismatch repair provides another mechanistic explanation for the higher mutations rates in closed chromatin (in addition to transcription-coupled DNA repair)”

Line 434: added “mutational”

Line 474: changed “but is also” to “but also how it is”

Line 488: changed “single cell RNA seq” to “single cell RNA sequencing"